# Efficiency and Persistence of Movento^®^ Treatment against *Myzus persicae* and the Transmission of Aphid-Borne Viruses

**DOI:** 10.3390/plants10122747

**Published:** 2021-12-13

**Authors:** Thomas Armand, Luâna Korn, Elodie Pichon, Marlène Souquet, Mélissandre Barbet, Jean-Laurent Martin, Magalie Devavry, Emmanuel Jacquot

**Affiliations:** 1PHIM Plant Health Institute Montpellier, University of Montpellier, INRAE, CIRAD, Institut Agro, IRD, CEDEX 5, 34398 Montpellier, France; thomas.armand@inrae.fr (T.A.); luana.korn@inrae.fr (L.K.); e.pichon90@hotmail.fr (E.P.); marlene.souquet@inrae.fr (M.S.); melissandre.barbet@outlook.fr (M.B.); 2Bayer S.A.S./Bayer CropScience, 16 rue Jean Marie Leclair, CS 90106, CEDEX 09, 69266 Lyon, France; jeanlaurent.martin@bayer.com (J.-L.M.); magalie.devavry@bayer.com (M.D.)

**Keywords:** aphicid treatment, *Myzus persicae*, plant-virus-aphid interactions, *Polerovirus*, beet mild yellowing virus, turnip yellows virus

## Abstract

Neonicotinoids are widely used to protect fields against aphid-borne viral diseases. The recent ban of these chemical compounds in the European Union has strongly impacted rapeseed and sugar beet growing practices. The poor sustainability of other insecticide families and the low efficiency of prophylactic methods to control aphid populations and pathogen introduction strengthen the need to characterize the efficiency of new plant protection products targeting aphids. In this study, the impact of Movento^®^ (Bayer S.A.S., Leverkusen, Germany), a tetrameric acid derivative of spirotetramat, on *Myzus persicae* and on viral transmission was analyzed under different growing temperatures. The results show (i) the high efficiency of Movento^®^ to protect rapeseed and sugar beet plants against the establishment of aphid colonies, (ii) the impact of temperature on the persistence of the Movento^®^ aphicid properties and (iii) a decrease of approximately 10% of the viral transmission on treated plants. These observations suggest a beneficial effect of Movento^®^ on the sanitary quality of treated crops by directly reducing primary infections and indirectly altering, through aphid mortality, secondary infections on which the spread of disease within field depends. These data constitute important elements for the future development of management strategies to protect crops against aphid-transmitted viruses.

## 1. Introduction

One of the most important challenges for a virus is to optimize its transfer between hosts. Thus, plant viruses have developed different strategies to make it possible to be transferred from infected to healthy hosts. Indeed, plant-to-plant transfer of viral particles can be achieved by contact between two plants (e.g., *Tobacco mosaic virus*, genus *Tobamovirus* [1]), by vegetative multiplication of infected plants (e.g., *Grapevine leafroll-associated virus-3*, genus *Ampelovirus* [2]), and/or by vertical transmission through the seeds (e.g., *Zucchini yellow mosaic virus*, genus *Potyvirus* [3]). However, for numerous viral species, host change depends on an organism which acts as vector (e.g., nematodes (e.g., *Grapevine fanleaf virus*, genus *Nepovirus* [4]), fungi (e.g., *Beet necrotic yellow vein virus*, genus *Benyvirus* [5]) or insects (e.g., *Wheat dwarf virus*, genus *Mastrevirus* [6])). These vectors can be considered as an advantage in the spread of viral pathogens. Indeed, vectors allow the virus (i) to reach new hosts located far from the viral source (i.e., from few meters until more than a kilometer [7]) and/or (ii) to be maintained outside host plant for long periods (i.e., from days to years [8]). However, vector-borne viruses that do not use alternative transmission strategies (e.g., *Beet chlorosis virus*; genus *Polerovirus* [9]) depend on the behavior and the biology of their vector(s) to escape from infected hosts [10,11]. Control strategies targeting vectors of viral diseases were developed at the end of the 20th century. For insect-borne viral disease, insecticide (e.g., neonicotinoids, pyrethoids, and/or carbamates) were widely used, especially for crops that did not have genetic resources to control viral infections. However, these chemicals can be associated to (i) toxic effects on non-targeted organisms [12,13] or (ii) the selection of insecticide-resistant vectors [14,15]. Consequently, some chemical families (e.g., neonicotinoids) have been banned in Europe [16], whereas other insecticides have lost their efficiency (emergence of resistant insects), making them obsolete for the control of vectors within fields [17,18].

Sugar beet (*Beta vulgaris subsp. vulgaris*) and rapeseed (*Brassica napus*) are exposed to numerous biotic stresses, of which viral yellow diseases caused on beet by different viruses [19] including *Beet mild yellowing virus* (BMYV, family *Solemoviridae*, genus *Polerovirus*) and on rapeseed by *Turnip yellows virus* (TuYV, family *Solemoviridae*, genus *Polerovirus*) [20,21]. These plant viruses, transmitted by aphids, mainly by the green peach aphid, *Myzus persicae* (Sulzer 1776) [22,23], cause important yield losses with up to 40% in case of early infections. For several decades, the main strategy to control these diseases in fields was based on the use of seeds coated with neonicotinoids (NNI) that killed aphids while they fed on treated plants. In the late 2010s, 98% of sugar beets and more than 25% of rapeseeds sown in France were treated with NNIs [24]. This mainly relies on (i) the high efficiency of chemicals from this insecticide family and (ii) the absence of resistance and/or tolerance to BMYV in sugar beet germplasm and description of only few partial resistance sources in rapeseed germplasm [25,26]. Due to the recent ban of NNI, any new chemical solution should be studied carefully in order to define the conditions of use that would allow it to provide efficient protection of plants while limiting the selection of resistant vectors.

Movento^®^ [Bayer CropScience], an ambimobile insecticide developed a decade ago, is a tetrameric acid derivative of spirotetramat. During its development, Movento^®^ showed high performance against insects (whiteflies, scales, mealybugs, psyllids, thrips, and aphids). This active ingredient acts as an inhibitor of lipid biosynthesis, impacts the development of larvae, and alters the fecundity of adults [27]. The application of Movento^®^ to plants allows it to penetrate the host tissues through the leaf cuticle. Once in the plant tissue, the spirotetramat is rapidly transformed into an active form and migrates in the whole plant via the phloem and the xylem. This mode of action allows it to reach insects present on all parts of the plant including surfaces not exposed to insecticide sprays (i.e., lower side of leaves and roots) [28]. Previous studies assessed efficiency and persistence of Movento^®^ on survival and sexual reproduction of *M. persicae* settled on several plant species including cabbage, pepper, and peach [27,28,29,30]. However, the impact of (i) the temperature on efficiency and persistence of Movento^®^ and (ii) Movento^®^ treatment on the efficiency of aphid-mediated viral transmission are not documented in the literature.

Evaluation of the properties of chemical targeting aphids can be carried out using experimental procedures in which an aphid is transferred on a test plant previously treated with the product to be tested. The survival of the aphid is assessed by observing it and its possible offspring for several days. The use of viruliferous aphids in such experiments allows the analysis of the impact of the insecticide on the success of the aphid-mediated viral inoculation. Finally, the persistence of the insecticide treatment can be described through the evaluation of insecticide efficiency at different dates after treatment. In this study, this entire procedure, based on the use of viruliferous *M. persicae* aphids (at the L_2_/L_3_ larvae stages) and susceptible plants (rapeseed (cv. DK Exception) and sugar beet (cv. VY0)) treated with Movento^®^, was carried out at different growing temperatures. In addition to the evaluation of the impact of Movento^®^ on both the survival of *M. persicae* and the viral transmission rates, the main objective of this work was to evaluate the effect of temperature on efficiency and persistence of Movento^®^ in order to help growers to optimize their future strategies to limit the presence of aphids (and the virus they transmit) in their fields.

## 2. Results

### 2.1. Efficiency of a Movento^®^ Treatment against Aphids

Untreated control plants maintained in a growth chamber at 24 °C for 15 days after the transfer of a founder aphid showed that these aphids were able to survive and produce of a small colony in 4/6 sugar beet plants (66.7%) and 10/10 rapeseed plants (100%) (Figure 1A, 24 °C). Under our experimental conditions, founder aphids released on test plants at the L_2_/L_3_ larvae stages can produce, in 15 days at 24 °C, colonies of 32.7 (+/−48.4) individuals on sugar beet cv. VY0 and 283.7 (+/−141.1) individuals on rapeseed cv. DK Exception (Figure 1B, 24 °C). The absence of aphids on the plants treated with Movento^®^ (treatments done 1 day before aphid transfer) shows that no founder aphids were able to survive on these plants (Figure 1A,B, 24 °C). Data generated under other thermal conditions (6 °C, 10 °C, 16 °C, and 20 °C) were collected and used to test the impact of temperature on the efficiency of Movento^®^ to prevent the production of aphid colonies. Data showed that the survival rate of aphids on untreated plants was 100%, with the exception of experiments carried out at 6 °C and 10 °C on sugar beet (only 5 and 0 of the 6 founder aphids survived on their untreated host plants) (Figure 1A, 6 °C and 10 °C). The number of aphids counted on untreated plants with colonies varied according to the temperature, ranging from 7.0 (+/−2.3) to 70.0 (+/−26.8) aphids/plant for sugar beet and from 24.8 (+/−10.5) to 283.7 (+/−141.1) aphids/plant for rapeseed (Figure 1B). For all temperatures, no colonies were observed on treated plants (Figure 1A,B), suggesting that in our conditions, Movento^®^ prevents the maintenance of aphids from 6 to 24 °C.

### 2.2. Persistence of Movento^®^ Protection against Aphids

To test the persistence of Movento^®^ properties against aphids, founder aphids (1 aphid per plant) were deposited on 6 untreated plants and on 10 Movento^®^-treated plants. at different dates after treatment (DAT), i.e., at 4, 8, 11, 14, 18, 21, and 25 days. The results obtained show that under some of the tested conditions (temperature and date of application of the founder aphid) plants treated with Movento^®^ allowed the establishment of a few individuals (from 1 to 5 aphids per plant) or the production of colonies (more than 5 individuals per plant). Thus, on rapeseed, first groups of individuals (number of aphids ≤ 5, Figure 2A, black circles) were observed for aphid transfers carried out at 25 DAT (experiment run at 16 °C), 18 DAT (experiment run at 20 °C), and 11 DAT (experiment run at 24 °C), whereas the first aphid colonies (number of aphid > 5, Figure 2A black triangles) were observed on rapeseed for aphid transfers carried out at 25 DAT (experiment run at 20 °C) and 11 DAT (experiment run at 24 °C). These observations make it possible to define the persistence of Movento^®^ efficiency for each temperature. The period during which the plant is protected from any aphid settlement corresponds to a time of full aphicid efficiency for Movento^®^ (Figure 2B, grey periods). A partial aphicid efficiency is observed when some of the treated plants allow the founder aphid to maintain itself and to produce few offspring (*n* ≤ 5; Figure 2B, white periods). The end of Movento^®^ aphicid efficiency is defined as a date from which the treatment can no longer alter the development and the production of aphid colonies (Figure 2B, black periods). Similar observations were obtained for the experiment carried out with sugar beet plants (Figure 2A, grey symbols). Sugar beet plants treated with Movento^®^ exhibit full, partial, and no protection according to the considered DAT and growing temperature. Full aphicid protection lasts at least 25 days (for the 6 °C and 10 °C conditions) and periods of partial protection start from 18 DAT (at 16 °C), 21 DAT (at 20 °C), and 4 DAT (at 24 °C). The end of aphicid efficiency on sugar beets were observed at 25 DAT (at 20 °C) and 18 DAT (at 24 °C) (Figure 2B). The 6 untreated plants included in each DAT/temperature combination were used as controls to monitor the establishment of aphid colonies in the absence of Movento^®^ treatment. For DAT/temperature combinations with full aphicid protection on treated plants, untreated control plants, used to confirm the viability of manipulated *M. persicae*, allowed establishment of 91.1% (82/90) and 83.9% (141/168) of aphids for rapeseed and sugar beet, respectively (not illustrated).

### 2.3. Impact of Movento^®^ on Viral Infection

The sanitary status (healthy vs. infected) of all plants used in the experiments was assessed using ELISA (enzyme-linked immuno-sorbent assay) tests (Table 1). The inoculation of viruses transmitted by aphids in a persistent manner requires that viruliferous aphids feed on the plants and therefore have the possibility to inoculate the virus in the phloem of hosts [31]. To test the impact of Movento^®^ on the aphid-mediated inoculation of BMYV to sugar beet and TuYV to rapeseed, temperature/DAT combinations for which observations did not show a full aphicid efficiency of Movento^®^ were removed from the data set (Table 1, non-bold combinations). Thus, the transmission rate observed for untreated plants was 69.1% (+/−7.1%) for sugar beet and 80.0% (+/−8.3%) for rapeseed, whereas treated plants were infected with an efficacy of 58.5% (+/−5.9%) for sugar beet and 68.0% (+/−7.5%) for rapeseed, highlighting a significant difference of infection rates (+/−10%) for treated plants (*p* = 0.027 for sugar beet and *p* = 0.039 for rapeseed) (Figure 3).

### 2.4. Behavior of M. persicae on Treated/Untreated Plants

To measure the effect of a treatment on the behavior of *M. persicae*, pots containing two treated sugar beets, two untreated sugar beets, or a combination of one treated and one untreated sugar beet were produced. One aphid was deposited per pot and pots were observed 1 and 5 days later. This make it possible to analyze the host choice made by an aphid according to the plants to which it can have access.

Observations carried out 1 day after aphids were deposited between the plants (Figure 4, left panels) show that in the presence of two untreated sugar beet plants, 96.7% of aphids (58/60, Figure 4A) succeeded in settling on one of the two plants. This indicates that under our experimental conditions, the mortality rate following the manipulation of larvae is very low (i.e., 3.3%). When the experiment was carried out with pots containing two treated plants, the mortality observed 1 day after the beginning of the experiment was 41.7% (25/60, Figure 4B). In the experiment carried out with two treated or two non-treated plants, the aphid has to choose between two equivalent hosts. Thus, no differences were noticed between the two plants for the colonization rates. In the experiment carried out with pots containing treated and non-treated hosts, the average colonization rates observed 1 day after the deposition of aphids were 27% (17/63) for the treated plants and 36.5% (23/63) for the untreated plants (Figure 4C). Furthermore, the mortality rate observed for pots with treated/untreated plants was 36.5% (23/63) (Figure 4C), which is close to the mortality rate observed for experiments carried out with pots containing two treated plants. Observation of plants on the fifth day of the experiment (Figure 4, right panels) showed (i) an increase in the mortality rate for untreated pots (i.e., 8/60 or 13.3%) and (ii) a pot with aphids observed on both plants (Figure 4D). For pots with two treated plants (Figure 4E), no live aphid was observed on the fifth day of experiment. Finally, in pot with both treated and untreated plants (Figure 4F), observations made on the fifth day of the experiment showed 3.2% (2/63) of live aphids on treated plants and a mortality rate of 65.1% (41/63). As each pot was individually observed at the first and the fifth day of the experiment, it is possible to estimate the survival rate of aphids at day 5 according to the host where they decided to settle at day 1 of the experiment. On pots with 2 treated plants, the aphids observed at day 1 were not able to survive until day 5 of the experiment (Figure 5A, treated pots). In contrast, 90% (+/−4.72%) of aphids observed at day 1 on an untreated plant from pots containing untreated plants only were alive at day 5 of the experiment (Figure 5A, untreated pots). In mixed environments, where the aphids had access to both treated and untreated plants, the data showed that 13% of the aphids initially observed on treated plants survived at day 5 (Figure 5B, treated plant), whereas 76% of the aphids that settled on untreated plants at day 1 were still alive on day 5 (Figure 5B, untreated plant).

## 3. Discussion

One of the main current challenges for rapeseed and sugar beet productions is to develop new strategies to control *Myzus persicae* and the viruses they transmit. Indeed, since the end of neonicotinoids (NNI) in Europe in 2018 [32], the control of aphid-borne viral diseases can no longer rely on the use of this family of insecticides. Due to the existence of *M. persicae* clones resistant to the other chemical families widely used to remove insects from cultivated fields (e.g., pyrimicarb and pyrethroids [15]), these insecticides cannot constitute an alternative to NNI. In the absence of efficient genetic and/or prophylactic solutions against yellows viruses, any new product should be tested to evaluate its potential for future development of control strategies. Protection of a field by insecticide can be achieved in a curative (after the observation of aphids in field) or in a preventive (before the arrival of insects) manners. Monitoring the arrival of aphids and their presence in the field is a time-consuming and unreliable activity. The time needed to carry out regular observation of plants (to look for the presence of aphids), the capacity of aphids to multiply rapidly (e.g., in a day, several nymphs are produced by an adult *M. persicae* [33]), and the ability of viruliferous aphids to persistently transmit viruses for several days [31] lead to situations where aphids, once observed in the field, have already had the opportunity to inoculate viruses to a significant proportion of plants before curative insecticide sprays can be applied. Moreover, treatments, whether curative or preventive, have a persistence that should be taken into account to protect crops during the entire period of exposure to the viral risk. Considering these characteristics, farmers could favor preventive treatment(s) to secure their crops. However, the application of insecticides as preventive management strategies induces the use of chemicals without considering either the intensity of aphid flights or the sanitary status (viruliferous or virus free) of the insects. This obviously leads to the use of more chemicals than would have been necessary to limit the incidence of aphid-borne viral disease and could promote the emergence of insecticides-resistant vectors.

The evaluation of Movento^®^ showed that, one day after a treatment, this chemical prevents insects from carrying out their biological cycle. The insecticide property of Movento^®^ is maintained for a period ranging from 11 to 25 days on rapeseed cv. DK Exception and from 4 to 25 days on sugar beet cv. VY0. The persistence observed under controlled conditions is temperature dependent. Indeed, increasing the temperature from 6 °C to 24 °C leads to a decrease in the period of protection against aphids. Young plants are more susceptible to virus infection than mature plants [34]. During the period of exposure, it is therefore important to protect plants against high virus risk (e.g., period of flight of aphid vectors of viral yellow diseases [16]) and more particularly for the early developmental stages of the plant. The growing season of rapeseed and sugar beet differ, among other things, by their sowing period. Indeed, rapeseed is sown in early autumn (a period when temperatures tend to decrease as autumn passes and winter approaches). Sugar beet, sown in early spring, grows in an environment where the temperature gradually increases until summer. The protection of these crops against the establishment and outbreak of *M. persicae* could require several treatments to protect plants throughout the entire period of exposure to the viral risk. Data acquired on the impact of temperature on the persistence of Movento^®^ treatments against aphids would allow adapting the frequency of insecticide treatments according to growing temperature. This would lead to progressively extend the period between two successive treatments for rapeseed (in conjunction with the decrease in the temperature from autumn to winter) and progressively bring two successive treatments closer together on sugar beet (in conjunction with the increase in the temperature during spring). The transfer of the information acquired under controlled conditions to fields would require refining persistence of Movento^®^ treatments according to the variations of temperature during days and nights in growing locations. This would make it possible to optimize delays between two applications in sugar beet and rapeseed fields. This study does not allow us to determine the maximum persistence of Movento treatment under low temperature conditions (i.e., 6 °C and 10 °C), as the experimental design did not test persistence longer than 25 days. However, in French fields, rapeseed and sugar beet growing periods with both viral exposure and low temperature (below 10 °C) do represent less than 25 days. This suggests that a single Movento^®^ treatment would efficiently protect rapeseed and sugar beet during low temperature growing periods.

It is important to note that, despite the use of Movento^®^, plants will be exposed to the inoculation of viruses (primary inoculations) by aphids landing on the treated area with the same intensity as for untreated fields. However, infection rates observed for plants protected by Movento^®^ were significantly lower (approximately 10% for both plant species). The impact of Movento^®^ on infection rate could be the result of interactions between treatment and virus, or more likely be a consequence of the aphicid property of Movento^®^. The experimental design used in this study does not allow us to test these two hypotheses. The 10% decrease in infected plants may contribute to the improvement of the sanitary quality of the protected areas by reducing the number of virus introductions in the field. In studies carried out on other annual plant species (e.g., cereals, potato), it was shown that the overall incidence of insect-transmitted viruses relies mainly on the efficiency of secondary infections initiated from the primary infected plants [35,36]. Thus, by preventing aphids from settling on treated plants and producing their offspring, and by lowering the success of primary inoculations, Movento^®^ may negatively impact important steps of viral disease spread at the field level.

When an aphid reaches a plant, several parameters linked to plant-aphid interactions allow the insect to initiate feeding and to start the production of a colony [37]. If some of these parameters are modified for Movento^®^ treated plants, the behavior of aphids could be impacted in different manners, including a possible modification of aphid mobility on and between plants. In such a scenario, the contribution of Movento^®^ to limit introductions and spread of aphid-transmitted viruses in fields would be lowered by an increase in the number of plants visited by aphids. Consequently, each aphid would have the possibility to feed and to inoculate virus to a higher number of plants before being killed by the insecticide treatment. Observation of the behavior of *M. persicae* with the simultaneous presence of one treated and one untreated sugar beet plants (‘mixed’ pots) indicated that the phytosanitary status of the host plant did not affect the choice made by the aphid to settle on one of these two types of plants (42.5% vs. 57.5% for treated and untreated plants, respectively). However, whereas individuals having access only to treated plants died before the fifth day of the experiment, living aphids were observed on the treated plants of the ‘mixed’ pots at the end of the experiment suggesting that a phenomenon, that remains to be characterized, extend the survival period (for at least 5 days) of aphids settled on a plant treated with Movento^®^ in the vicinity of an untreated plant.

This study provides information about the aphicid efficiency and persistence of Movento^®^, by indicating, under different temperature conditions, the optimum delays between two treatments to maintain a high level of insecticide protection and a low incidence of viral disease. With this information, sugar beet and rapeseed growers will have the possibility to optimize direct (application(s)) and indirect (potential impact on the environment) costs of a Movento^®^-based management strategy.

## 4. Materials and Methods

### 4.1. Plants, Aphids, and Viruses

Rapeseed cv. DK Exception and sugar beet cv. VY0 were used in the experiments for their ability to produce aphid colonies and their susceptibility to Turnip yellows virus (rapeseed) and Beet mild yellowing virus (sugar beet). Rapeseed and sugar beet were sown on N2 soil (Neuhaus^®^ Huminsubstrat N2, Klasmann Deilmann, Geeste, Germany) in trays (L ×W × H: 13.4 × 12.2 × 4.9 cm) and maintained in a growth chamber (day/night: 12 h/12 h, 23 °C/20 °C). Seedlings were transplanted 5–7 days after sowing into individual pots ( Pöppelmann TEKU^®^ [Rixheim, France], 7 × 7 cm). After transplanting, the pots were placed in a greenhouse (day/night, 24 °C/20 °C) for 7 days before being used in the experiments.

*Myzus persicae* clone Mp34 [38] was reared in a growth chamber (day/night: 16 h/8 h, 24 °C/20 °C, RH: 40%) in plexiglass cylinders in the presence of healthy rapeseed or sugar beet plants. Isolates PS of Turnip yellows virus (TuYV-PS; [26]) and itb2 of Beet mild yellowing virus (BMYV-itb2; [20]) were maintained in collection in the presence of Mp34 aphids on rapeseed and sugar beet plants, respectively.

### 4.2. Viruliferous Aphids to Inoculate Plants

To evaluate the efficiency and persistence of a Movento^®^ treatment against *M. persicae*, pots (7 × 7 cm, N2 soil) of rapeseed or sugar beet (one plant per pot) were prepared. The plants were treated by spraying 126 µL/pot of a Movento^®^ solution (commercial solution diluted 1/572 (*v*/*v*) in order to reproduce the dose for field treatments, according to manufacturer’s recommendations). Untreated plants were used as controls. The day after the treatment, groups of 6 untreated and 10 treated plants were constituted for each plant species and transferred to a growth chamber, allowing the experiment to be conducted at 6 °C, 10 °C, 16 °C, 20 °C, and 24 °C. For the experiments conducted at 6 °C and 10 °C, only sugar beet plants were used. On each plant of a group, a viruliferous (Mp34/TuYV-PS for rapeseed plants and Mp34/BMYV-Itb2 for sugar beet plants) *M. persicae* (L_2_/L_3_ larvae stages) was deposited with a brush on the main stem of the plant. This step was done with the different groups of plants at eight dates after treatment (i.e., at 1, 4, 8, 11, 14, 18, 21, and 25 days after treatment (DAT)). Pots were then individually covered with a micro-perforated plastic bag to confine the aphid and the plant for 15 days. At the end of the experiment, aphid population present on each plant was counted. After counting, plants (leaves and stem) were individually sampled and stored at −20 °C until the analysis of their sanitary status (i.e., presence of TuYV (rapeseed) or BMYV (sugar beet)) by ELISA tests.

### 4.3. Plant/Aphid Interactions

The impact of a Movento^®^ treatment performed on young plants on the behavior of *M. persicae* was tested using treated and untreated plants. Pots (7 × 7 cm, N2 soil) of sugar beet cv. VY0 (2 plants per pot) were prepared. Seven days after sowing, some pots were treated by spraying 126 µL/pot of a Movento^®^ solution (commercial solution diluted 1/572 (*v*/*v*)), whereas other pots were not treated. To prepare pots with one treated plant and one untreated plant, a transplanting step (replacement of an untreated plant by a treated plant in untreated pots) was carried out 24 h after treatment. A viruliferous (BMYV-itb2) *M. persicae* (L_2_/L_3_ larvae stages) was then deposited in the centre of each pot on an area previously covered with a small piece of paper (1 cm²) to facilitate first movement (post-deposition) of the aphid to reach plant(s). The pot was then covered with a micro-perforated plastic bag to confine the aphid and the two plants for the duration of the experiment. One and five days later, each plant was observed in order to identify which plant(s) had been colonised by the aphid (and its offspring). After the last observation, the plants were treated with an insecticide (Pirimor, 0.1% (*v*/*v*); Syngenta, Basel, Switzerland) and kept for 21 days in the greenhouse before being tested for the presence of BMYV by ELISA. The whole experiment was repeated three times.

### 4.4. Serological Detection of Viruses in Plant Samples

Individual plants (leaves and stems) were collected and stored at −20 °C until serological analysis. This biological material was ground with a Pollähne press (MEKU, Wennigsen, Germany). The resulting plant sap was used for virus detection by enzyme-linked immuno-sorbent assay (ELISA) [39]. In each well of a microtiter plate (NUNC, Maxisorp, Denmark), 100 µL of an antibody solution diluted according to the manufacturer’s recommendations (1/200 for IgG-TuYV antibodies (LOEWE, Sauerlach, Germany) and 1/1000 for IgG-BMYV antibodies (DSMZ, Braunschweig, Germany) in carbonate buffer (15 mM Na_2_CO_3_, 35 mM NaHCO_3_, pH = 9.6) was applied. Adsorption of the antibodies onto the plastic of the wells was performed for 4 h (TuYV) or 3 h (BMYV) at 37 °C. Between each ELISA step, the plates were washed three times with PBS-T buffer (137 mM NaCl, 8 mM Na_2_HPO_4_, 12H_2_O, 2.7 mM KCl, 1.5 mM KH_2_PO_4_, 0.05% (*v*/*v*) Tween 20) using the Biostack microplate washer (BIOTEK, Bruchsal, Germany). For the detection of BMYV by TAS-ELISA, the wells were saturated (30 min at 37 °C) with PBS-T buffer supplemented with 2% (*w*/*v*) skimmed milk before the samples were deposited in wells. After deposition of 100 µL of plant sap in each well, the microtiter plate was placed overnight at 4 °C. A diluted alkaline phosphatase (@-PA) coupled antibody (TuYV: 1/200; BMYV: 1/1000) in conjugate buffer (PBST buffer, 2% (*w*/*v*) polyvinylpyrrolidone 40T, 2% (*w*/*v*) ovalbumin) was deposited and incubated for 4 h (TuYV) and 1 h (BMYV) at 37 °C. For the detection of BMYV, an additional step consisting of incubating the plate for 3 h in the presence of a secondary antibody diluted 1/1000 in conjugate buffer was performed before the addition of the @-PA antibody. Finally, wells were filled with 100 µL of substrate buffer (diethanolamine (1 N, pH = 9.8)) containing para-nitrophenylphosphate (PNPP) (1 mg/mL). The incubation was done at room temperature in the dark. After 1h, the intensity of the colorimetric reaction was measured at 405 nm (OD_405_) for each well using a spectrophotometer (Multiskan FC; Thermo Scientific, [Waltham, MA, USA]). The threshold for assay positivity was set at twice the OD_405_ value of the healthy controls with a minimum value of OD_405_ = 0.1.

### 4.5. Statistical Analyses

Statistical analyses were performed using R version 3.6.0 [40]. Binary data (i.e., infected/healthy plants or infested/non-infested plants) were analyzed using a binomial generalized linear model (GLM) followed by one-way ANOVA. Factor included in variance analyses is “treatment”. The effects were tested by comparing nested models using a Student test (drop1 function).

## Figures and Tables

**Figure 1 plants-10-02747-f001:**
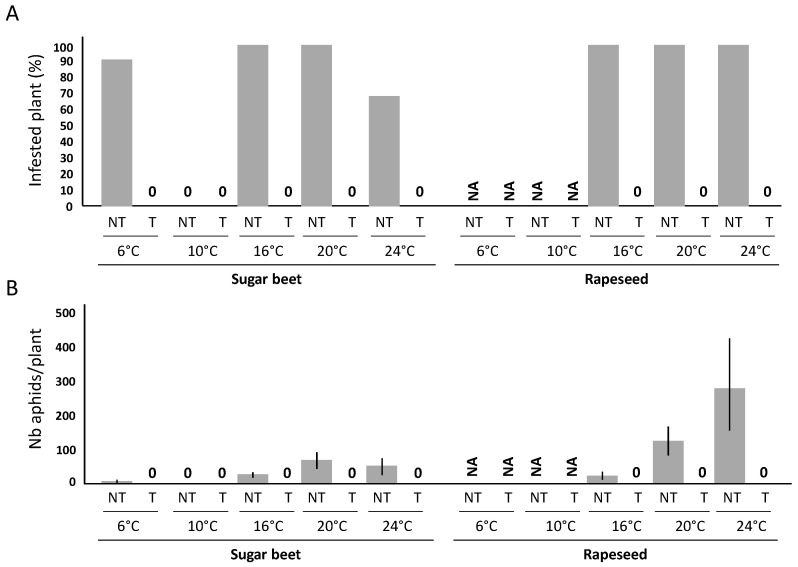
Survival rate and colony production of *M. persicae* on sugar beet and rapeseed. Sugar beet (cv. VY0) and rapeseed (cv. DK Exception) plants that have received (T) or not received (NT) a Movento^®^ treatment were used one day after the treatment (DAT 1) to host a founder *M. persicae* L_2_/L_3_ larvae for a period of 15 days. At the end of this period, the percentage of plants with live aphid(s) is recorded (**A**). The size of the aphid populations on the infested plants is counted (**B**). The black bars indicate the standard deviations associated with the observed values for aphid population sizes. Experiments were carried out at different temperatures (i.e., at 6, 10, 16, 20, and 24 °C). For each temperature condition, 10 inoculated plants and 6 control were used. NA: data not available.

**Figure 2 plants-10-02747-f002:**
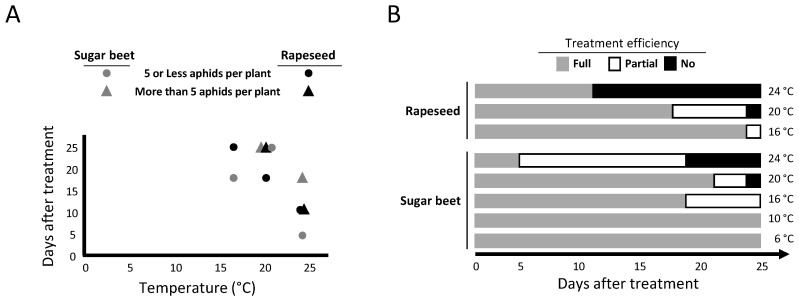
Persistence of Movento^®^ treatment. Plants of rapeseed cv. DK Exception and sugar beet cv. VY0 treated with a solution of Movento^®^ were maintained under different thermal conditions (6, 10, 16, 20, and 24 °C). After adding a *M. persicae* on plant at different dates after treatment (from 1 to 25 days), the survival of the aphid and its ability to produce Few larvae (observation of up to 5 individuals on a plant) or a colony (observation of more than 5 individuals on a plant) were observed (**A**). For each temperature tested, persistence of Movento^®^ is illustrated by the delay (days after treatment) required to allow the settlement of an aphid on treated plants and the production of few individuals (circles) or a colony (triangles). The presence/absence of aphids on the treated plants at the end of the experiment makes it possible to define the duration of full, partial, or no protection of Movento^®^ against aphids (**B**).

**Figure 3 plants-10-02747-f003:**
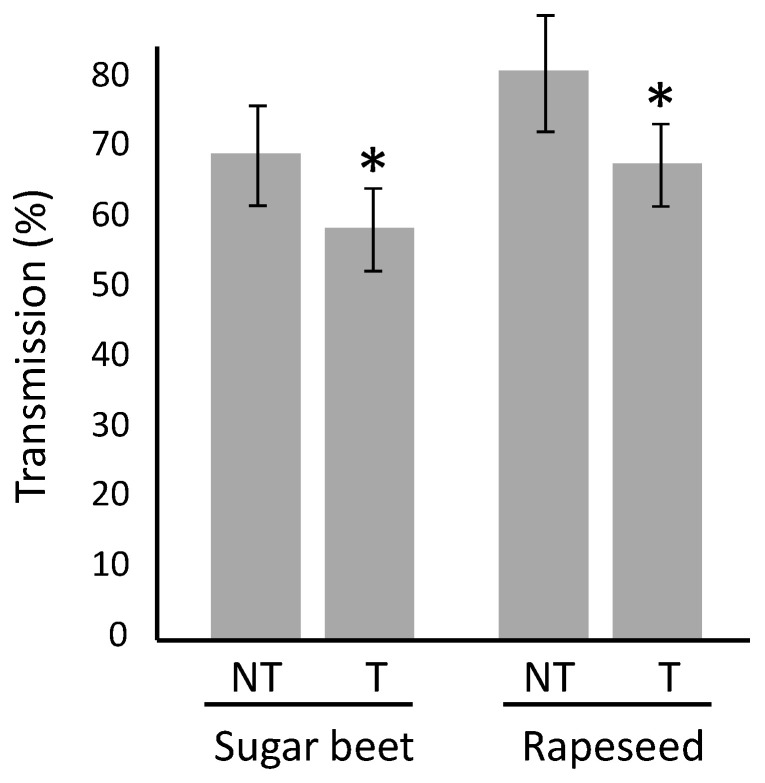
Infection rates for BMYV on sugar beet and TuYV on rapeseed. Sugar beet cv. VY0 and rapeseed cv. DK Exception, whether (T) or not (NT) protected by a Movento^®^ treatment, were exposed to virus infection by the transfer of a viruliferous aphid (L_2_/L_3_ larvae stage). After the maintenance of the viruliferous aphid on the test plant for a period of 15 days, the sanitary status of the plants was evaluated by an ELISA test. The black bars correspond to the standard deviations. Asterisks (*) indicate a significant difference (*p* < 0.05) between untreated and treated plants from a species (GLM binomial followed by a two-way ANOVA and HSD-Tukey post hoc test).

**Figure 4 plants-10-02747-f004:**
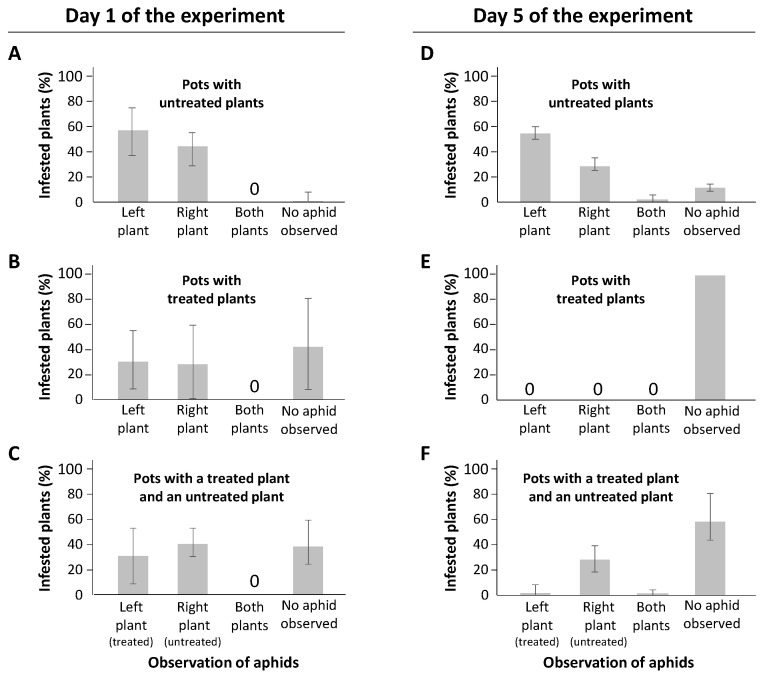
Observation of aphids on treated/untreated plants. Aphids (*M. persicae*, 1 aphid per pot) were deposited on pots between two treated plants (**A**,**D**), two untreated plants (**B**,**E**), or a combination of one treated plant (left) and one untreated plant (right) (**C**,**F**). After 1 day (**A**–**C**) and 5 days (**D**–**F**), the plants on which aphids had settled were identified. Pots where both (presence of an offspring) or any (no aphids observed) plants were infested were assigned to “both plants” and “No aphid observed”, respectively. Experiment was carried out with a series of 20 pots/combination of treated/untreated plants. The experiment was conducted three times. The error bars represent standard deviation.

**Figure 5 plants-10-02747-f005:**
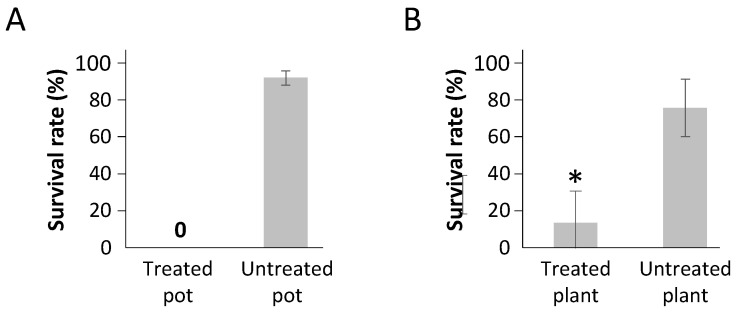
Aphid survival rate. Plants from treated and untreated pots (**A**) and from mixed pots (containing one treated and one untreated plant) (B) were observed for the presence of live aphids on day 1 and on day 5 of the experiment. The graphs illustrate the survival rates for treated and untreated pots (**A**) and for treated and untreated plants from mixed pots (**B**) at day 5 based on plants with live aphid at day 1 for each type of pot/plant. The black bars indicate the standard deviation (**B**). Asterisk (*) indicates a significant difference (*p* < 0.05) between untreated and treated plants (GLM binomial followed by a two-way ANOVA and HSD-Tukey post hoc test).

**Table 1 plants-10-02747-t001:** Infection rates of BMYV and TuYV on their respective treated (T)/untreated (NT) hosts according to day after treatment-temperature combinations.

		6 °C		10 °C		16 °C		20 °C		24 °C	
		T	NT	T	NT	T	NT	T	NT	T	NT
Sugar beet	D_1_	**3/10 (30%)**	**2/6 (33.3%)**	**1/10 (10%)**	**6/6 (100%)**	**10/10 (100%)**	**6/6 (100%)**	**10/10 (100%)**	**6/6 (100%)**	**7/10 (70%)**	**5/6 (83.3%)**
	D_4_	**4/10 (40%)**	**2/6 (33.3%)**	**8/10 (80%)**	**2/6 (33.3%)**	**9/10 (90%)**	**6/6 (100%)**	**9/10 (90%)**	**6/6 (100%)**	10/10 (100%)	6/6 (100%)
	D_8_	/	/	/	/	**10/10 (100%)**	**6/6 (100%)**	**10/10 (100%)**	**6/6 (100%)**	7/10 (70%)	4/6 (66.6%)
	D_11_	**5/10 (50%)**	**2/6 (33.3%)**	**8/10 (80%)**	**6/6 (100%)**	**8/10 (80%)**	**6/6 (100%)**	**8/10 (80%)**	**5/6 (83.3%)**	6/10 (60%)	5/6 (83.3%)
	D_14_	**10/10 (100%)**	**1/6 (16.7%)**	**7/10 (70%)**	**2/6 (33.3%)**	**4/10 (40%)**	**6/6 (100%)**	**5/10 (50%)**	**6/6 (100%)**	6/10 (60%)	5/6 (83.3%)
	D_18_	**0/10 (0%)**	**0/6 (0%)**	**1/10 (10%)**	**6/6 (100%)**	8/10 (80%)	6/6 (100%)	**7/10 (70%)**	**4/6 (66.6%)**	3/10 (30%)	4/6 (66.6%)
	D_21_	**2/10 (20 %)**	**4/6 (66.6%)**	**4/10 (40%)**	**6/6 (100%)**	8/10 (80%)	6/6 (100%)	**2/10 (20%)**	**1/6 (16.7%)**	9/10 (90%)	3/6 (50%)
	D_25_	**0/10 (0%)**	**3/6 (50%)**	**6/10 (60%)**	**1/6 (16.7%)**	7/10 (70%)	5/6 (83.3%)	9/10 (90%)	6/6 (100%)	5/10 (50%)	0/6 (0%)
Rapeseed	D_1_	/	/	/	/	**8/10 (80%)**	**5/6 (83.3%)**	**8/10 (80%)**	**5/6 (83.3 %)**	**7/10 (70%)**	**6/6 (100%)**
	D_4_	/	/	/	/	**9/10 (90%)**	**6/6 (100%)**	**9/10 (90%)**	**5/6 (83.3%)**	**4/10 (40%)**	**1/6 (16.7%)**
	D_8_	/	/	/	/	**8/10 (80%)**	**5/6 (83.3%)**	**8/10 (80%)**	**5/6 (83.3%)**	**4/10 (40%)**	**4/6 (66.6%)**
	D_11_	/	/	/	/	**9/10 (90%)**	**5/6 (83.3%)**	**7/10 (70%)**	**6/6 (100%)**	4/10 (40%)	3/6 (50%)
	D_14_	/	/	/	/	**4/10 (40%)**	**6/6 (100%)**	**6/10 (60%)**	**5/6 (83.3%)**	3/10 (30%)	1/6 (16.7%)
	D_18_	/	/	/	/	**5/10 (50%)**	**4/6 (66.6%)**	8/10 (80%)	4/6 (66.6%)	2/10 (20%)	1/6 (16.7%)
	D_21_	/	/	/	/	**6/10 (60%)**	**4/6 (66.6%)**	5/10 (50%)	2/6 (33.3%)	7/10 (70%)	1/6 (16.7%)
	D_25_	/	/	/	/	9/10 (90%)	6/6 (100%)	3/10 (30%)	3/6 (50%)	7/10 (70%)	3/6 (50%)

T: Plants treated with Movento^®^. NT: Untreated plants. D_x_: Day X after treatment. /: combination not tested. Data are presented as number of infected plants/number of tested plants for each combination. Results presented in bold correspond to combinations with efficient Movento^®^ protection against aphids.

## Data Availability

Not applicable.

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
