# Peer review of "Efficiency and Persistence of Movento® Treatment against Myzus persicae and the Transmission of Aphid-Borne Viruses"

_plants, 2021, doi:10.3390/plants10122747_

Round 1

Reviewer 1 Report

The manuscript “Efficiency and persistence of Movento® treatment against Myzus persicae and the transmission of aphid-borne viruses” by Th. Armand et al. presents original results on the potential of Movento in its use as an aphid-cide against polerovirus-induced diseases in sugarbeet and rapeseed.  The experimental settings are well done and the interpretation of the results is sound.  They bring interesting information for a future integration of Movento in control strategies, even though there is always a gap to fill between lab work and effectiveness in field crops.  Finally, the written style is sometimes difficult to follow and the Ms should be sumitted to a native English speaker.

I do recommend this Ms for acceptance in Plants, provided the comments and specific points below are addressed by the authors.

Specific comments

Line 1 and elsewhere: ® or ™?

L 2: italicize Myzus persicae; idem (L 16-17, 34, 116, …)

L 26: persicae

L 34-35: Grapevine leafroll associated virus-3

L 51: neonicotinoids

L 52: other insecticides

L 56: also BChV, cited above

L 72: mealybugs

L 107-108: what about sugarbeet at 24°, also an “exception” to 100%.

L 109-110: survived and untreated (x 2)

L 114: prevents

L 124 et seq.: how many founders per plant? One as L 97, I presume. How many plants? Did you use untreated controls?

L 137: efficiency

L 145: lasts

Table 1: the caption is not explicit and I found it uneasy to understand. Suggestion: “Transmission rates of BMYV on … and TuYV on … using viruliferous M. persicae nymphs transferred for inoculation on treated or untreated recipient plants, according to treatment–temperature combinations.” or something of the sort.  Also, precise that D1, D2, etc. refer to DAT.

Fig. 3: could the significance level (*, **…) be added? (see L 167-168)

L 181: that one aphid per pot was deposited per pot (as mentioned in Mat&Met) should be recalled here.

L 194: please start a new paragraph with “Observations…”

L 195: showed

Fig. 4: perhaps replace “Any plant” by “Mortality”?  Does it include unsettled/lost aphids?

L 200: why “each (65.1%)”?

L 201: days

L 202-206: this sentence could be split into 2.

L 219: The caption must be rewritten.

L 229: do not capitalize active ingredients

L 237: nymphs … an adult female of… (nymphs are also female)

L 255-256: perhaps add a ref to this statement

L 257: closing bracket missing

L 279+281: somewhat awkward sentence.

L 282 et seq.:  the authors could mention that primary epidemics is mainly due to winged aphids arriving into crops, whereas their experiments were done with wingless aphids.  Are there any differences in settling behavior? and between lab experiments and natural conditions?

L 292: impact

L 312: aphicide or aphicidal

L 324: Klasmann

L 377: remove one closing bracket

L 390: what was the secondary antibody? Anti-rabbit or …?

Ref 4: why journal in German and in English?

Ref 19: remove ™

Ref 20 and 24: incomplete refs

Ref 34: why capitals?

Author Response

Manuscript ID : plants-1478458

Efficiency and persistence of Movento® treatment against Myzus persicae and the transmission of aphid-borne viruses

Information linked to the revision process:

In this document, the reviewer’s comments that request an answer from authors are presented in blue and author’s comments are in black.

This would help editor and reviewers to follow how authors have upgraded the manuscript to produce the revised version

Reviewer comments:

 # Reviewer 1 

Line 1 and elsewhere : ® or ™ ?

Done, it is « ® » ( https://www.bayer-agri.fr/produits/fiche/insecticides-movento/ )

L 2: italicize Myzus persicae; idem (L 16-17, 34, 116, …)

Done

L 26: persicae

Done

L 34-35: Grapevine leafroll associated virus-3

Done

L 51: neonicotinoids

Done

L 52: other insecticides

Done

L 56: also BChV, cited above

The sentence has been modified and a reference (Hossain et al., 2020; [19]) has been added to illustrate the role of different viruses in the expression of yellow symptoms on beet.”

L 72: mealybugs

Done

L 107-108: what about sugarbeet at 24°, also an “exception” to 100%.

The results for sugar beet at 24°C are presented at the beginning of the paragraph (L101) :

« Untreated control plants maintained in a growth chamber at 24°C for 15 days after the transfer of a founder aphid showed that these aphids were able to survive and produce of a small colony in 4/6 sugar beet plants (66.7%) and 10/10 rapeseed plants (100%) (Fig. 1A, 24°C). »

Manipulation of aphids (especially larvae) could lead (even with the careful use of a brush) to the death of transported individual. As experiments start with a single aphid (1 aphid per plant), some of them could die after their transfer to test plant, leading to the observation of aphid-free plant at the end of the experiment.

Our experiments carried out with untreated plants lead to 9 plants out of 80 (11.25%) with no aphid (aphid died after transfer). This is not an “exception” to 100%. It is the variability of our ability to transfer ‘live’ insects to plants.

L 109-110: survived and untreated (x 2)

Done

L 114: prevents

Done

L 124 et seq.: how many founders per plant? One as L 97, I presume. How many plants? Did you use untreated controls?

The number of funder aphids used in the experiment has been clearly mentioned at the beginning of paragraph 2.2. At each tested temperature, 6 untreated plants were used as control to evaluate the aphid survival and the production of offspring in the absence of insecticide (Movento®) treatment. This materiel correspond to the untreated control.

“ To test the persistence of Movento® properties against aphids, founder aphids (1 aphid per plant) were deposited on 6 untreated plants and on 10 Movento®-treated plants. at different dates after treatment (DAT), i.e. at 4, 8, 11, 14, 18, 21 and 25 days.”

L 137: efficiency

Done

L 145: lasts

Done

Table 1: the caption is not explicit and I found it uneasy to understand. Suggestion: “Transmission rates of BMYV on … and TuYV on … using viruliferous M. persicae nymphs transferred for inoculation on treated or untreated recipient plants, according to treatment–temperature combinations.” or something of the sort.  Also, precise that D1, D2, etc. refer to DAT.

The caption has been modified:

Fig. 3: could the significance level (*, **…) be added? (see L 167-168)

Figure 3 has been edited according to reviewer’s request

L 181: that one aphid per pot was deposited per pot (as mentioned in Mat&Met) should be recalled here.

Done

L 194: please start a new paragraph with “Observations…”

Done

L 195: showed

Done

Fig. 4: perhaps replace “Any plant” by “Mortality”?  Does it include unsettled/lost aphids?

This class of data includes cases where no aphids were identified at the end of the experiment. We initially used "Any plants" to describe observations (as aphid were observed on any plants), while mortality is the interpretation of this observation. However, following your comment, figure 4 has been edited as requested.

Yes, at that point it is not possible to distinguish between dead, lost and unsettled aphids. They all fit in the initial “Any plant”/now “No aphid observed” data.

L 200: why “each (65.1%)”?

The “each” has been deleted

L 201: days

Done

L 202-206: this sentence could be split into 2.

Done

L 219: The caption must be rewritten.

Done

L 229: do not capitalize active ingredients

Done

L 237: nymphs … an adult female of… (nymphs are also female)

The sentence has been edited:

L 255-256: perhaps add a ref to this statement

Panter and Jones (2002) [32] added.

L 257: closing bracket missing

Added

L 279+281: somewhat awkward sentence.

The sentence has been modified.

L 282 et seq.:  the authors could mention that primary epidemics is mainly due to winged aphids arriving into crops, whereas their experiments were done with wingless aphids.  Are there any differences in settling behaviour? and between lab experiments and natural conditions?

We do not have information about settling behaviour of winged and wingless aphids. Experiments have been carried out with wingless aphids for several technical and experimental reasons:

- The Mp34 clone is maintained under rearing conditions optimized for production of wingless aphids

- Winged (sexual) aphids produce less individuals than parthenogenetic wingless aphids and one objective of the study was to monitor the production of offspring

- using winged aphids, it would be difficult to run the experiment designed to monitor the behaviour of aphids on treated/untreated plant (choice of plants).

Study carried out under lab/controlled conditions does not reflect 100% of natural conditions. We conducted experiments under the described lab conditions and the presented results were always considered according to the experimental design (including its main characteristics, i.e. lab conditions...). Of course, in the discussion section, we try to discuss the results under wider conditions, i.e. from lab to field.

In the main text, we used 5 times the terms ‘under our experimental conditions’, to make reader aware that work has been conducted in the lab.

L 292: impact

Done

L 312: aphicide or aphicidal

Aphicid was used in the main text

L 324: Klasmann

Done

L 377: remove one closing bracket

Done

L 390: what was the secondary antibody? Anti-rabbit or …?

Characteristics of commercial antibodies are not always described by providers. As mentioned in the M&M section, antibody used in the study were provided by Loewe and DSMZ. We followed manufacturer’s recommendations to run detection tests. Information on the origin (organism) of antibodies used in serological test does not seem relevant for the analysis of the results. Positive and negative controls were introduced in the tests to qualify the assay. Tests gave expected results for controls and were accurately analysed.

Ref 4: why journal in German and in English?

Done

Ref 19: remove ™

Done

Ref 20 and 24: incomplete refs

As references have been inserted in the reference list of the submitted manuscript, reference #20 is now reference #21 and reference #24 is now reference #25.

About reference #21: it comes from ICTV web site and has been similarly cited by Walker et al. (2021) (https://link.springer.com/article/10.1007/s00705-021-05156-1 ). To complete the reference, we added the website link toward the ICTV proposal.

About reference 25: the meta data obtained from Pub Med data bases, i.e. authors, title, abbreviated journal name, year of publication, volume and DOI, were used with Plants guidelines for references. The page range is not available yet as this article has been printed in November 2021 (https://pubmed.ncbi.nlm.nih.gov/33487015/ ;  22/11/2021).

Ref 34: why capitals?   

As references have been inserted in the reference list of the submitted manuscript, reference #34 is now reference #38.

ISHS Acta Horticulturae ask to use capital letters to cite their publications. We agree to follow Plants guidelines (https://www.actahort.org/books/386/386_27.htm ; « How to cite thus article »). Thus, If refrence #38 must be transformed form capital letters to standard letters to be used in Plants, then we ask Plants editor to modify the original typo of this reference to Plants standards.

# Reviewer 2

In the introduction, previous studies about Movento® on aphids needs to be explicitly presented. What differences are between this study and the previous studies and what new information the current study present comparing to previous studies ?

Introduction of the manuscript has been completed with requested data

For all figures, indications on the corresponding statistics will be helpful for readers to easily catch the point of the figures.

Done

Figure 1A, error bars are missing, my understanding is that the experiment was only performed once, and at least the variations across different plants need to be presented.

Figure 1A illustrates results obtained from one experiment. Errors bars cannot be added because of the binary nature of the data. Indeed, plants (6 plants for untreated (NT) and 10 plants for treated (T)) are either infested (data value is 1) or not (data value =0). The presence of aphid on these plants allowed to calculate the percentage of infested plants. It is not possible to add (to calculate) standard error with this type of data sets.

Figure 2A, each symbol represents data from a single plant? It is not entirely clear what each circle or triangle represents?

Grey and black colour code illustrate sugar beet and rapeseed, respectively. Circles and triangles represent the observation on a plant of few aphids (<5) or of a colony, respectively. Finally, for each temperature/days after treatment combination, a schematic representation of the results is presented as a symbol (circles and triangles).

As an example : experiments carried out at 16 degree Celsius:

- on sugar beet, plants with few aphids (<5) were observed on plants treated 18 days earlier (18 days after treatment), a grey circle is presented

- on rapeseed, plants with few aphids (<5) were observed on plants treated 25 days earlier (25 days after treatment), a black circle is presented…

We have modified the legend to help readers in the understanding of this important figure.

Figure 2B, there was no control data presented. Without controls, some of the statement in the manuscript is not accurate. For example, the author stated that “Full aphicid protection last at least 25 days (for the 6 and 10 degree conditions)”, as the data from figure 1, in most of the cases, no aphids survival even on the non-treated plants at those temperatures. Thus, it is not convincing that the protection is coming from Movento®.

The figure illustrates the protection of Movento® against aphids. Each tested condition included control (untreated) plants to allow comparison of treated/untreated condition.

A sentence has been introduced in the main text to describe data obtained with control untreated plants.

Table 1: the data presented is rather preliminary. At least a conversion of the raw data into percentages to allow direct comparisons between different groups

Done.

Figure 3: The ELISA data needs to be presented. In addition, the statistics for the difference test needs to be presented. To my understanding, the differences in the virus transmission is mainly caused by the decreases in aphid, but not a direct cause from Movento® to virus infections, further discussion may be needed for this possibility.

Figure 3 illustrated transmission efficiency for plants protected by Movento® and for plants non protected by Movento®. To avoid misintermpretation of the effect of Movento® treatment on transmission efficiency, temperature/days after treatment combinations associated to a non-full aphicid protection were removed from the data set as explained in the following sentence ;

To test the impact of Movento® on the aphid-mediated inoculation of BMYV to sugar beet and TuYV to rapeseed, temperature/DAT combinations for which observations did not show a full aphicid efficiency of Movento® were removed from the data set (Table 1, unbold combinations). “

This allow to compare efficiency of aphid inoculation on untreated and Movento®-protected plants.

We do not conclude that the reduction of infection rates is the direct cause of Movento® on virus as we do not have any data that makes it possible to test/analyse this Movento®/virus interactions.

In the results section, we do only describe the observed data. Then, in the discussion section we only report the 10% decrease.

To avoid misinterpretation of our results, we have added a sentence in the discussion.

Figure 4: in the choice assays, it is just confusing by leaving out what does the “both plants” and “any plants” mean. And why can’t replace the “left vs Right plants” with “treated vs non-treated”? To be able to understand why one aphid could be found in Both plants or any plants, which confused me, more details probably need to be provided for how the choice assays were done.

- In the results section, it was important to be able to distinguished plants from pots with two identical plants (treated or untreated) because it was important to prove the absence of bias in the aphid choice according to the position of the plant in the pot/tray/growth chamber. Thus, we choose to describe the plant according to their location in the experimental design (i.e. right and left) and prove that aphid is not impacted by the plant position during plant choice

- Aphids can be found on both plants when at least a larvae was produce (and observed) during the experiment. This was indicated in the caption.

- Aphids not observed on plants could be dead or hidden in the pot, they were assigned to ‘Any plant’ to not exclude (in the text) the possibility that they are somewhere on the soil at the moment of the observation. We propose to change “Any plant” by “No aphid observed” in the figure.

We edited the legend of figure 4.

Figure 5: the data presented seems about the percentages of plants were colonized by aphids, however, the description is about aphid survival. These are two totally different things. Either the data needs to be replaced with correct data, or the explanation needs to be corrected to reflect what the data says.

Legend of the figure has been edited.

I also caught some typos or incomplete and/or confusing sentences, format inconsistencies, the authors need a read through for those minor things to be corrected.

The whole manuscript has been edited to correct all typos and confusing sentences.

# Reviewer 3

If temperatures of 10oC or below are rare, why were levels corresponding to 6o and 10o  included in the design?  Once included in the design, why were they only tested for sugarbeets and not rapeseed?

When the experiment was designed, the original idea was to test efficiency and persistence of Movento on Sugar beet and on Rapeseed to optimize their use on these crops at the period of aphid/virus exposure.

Before getting the results, we did not know how long could be the persistence of Movento® treatment (days, weeks, months…) for each temperature. Thus, it was necessary to decide temperature to be tested and length of possible persistence.

For the temperatures, we consider that period of aphid/virus exposure in rapeseed is from sowing (September) to cold winter (i.e. late November/early December). This gave us a temperature window from 24°C to 0°C, when all aphids die. For Sugar beet (sown in march), temperature of exposure start from few degree above 0°C (especially because sugar beets are grown in the north/NorthWest of France where some winters are warm and could let aphids alive during the whole winter) to the summer were temperature reach 24°C (and sometimes warmer).

Thus, we decided to :

- test low (from 6°C) to high (24°C) temperature conditions…

But we also consider that low temperatures in rapeseed production is at the end of aphid/virus exposure (as low temperatures come with the beginning of winter) and insecticide use for Rapeseed in November (few days before the cold period of winter) sounds useless for growers. For sugar beet, low temperatures are at the beginning of aphid/virus exposure. Thus, insecticide treatment under low temperature sounds an important option to protect this crop.

Figure 1: why didn't aphids establish on sugarbeet at 10o, but did establish at 6o or above 10o?

We do not know.

We did not use this surprising result in the data analysis (just reporting it in results section). Data associated to persistence of Movento® allowed to test establishment and production of colonies of M. persicae at 10°C (using control plants together with treated plants at different days after treatment (from 1 to 25 days))…and on control plants maintained at 10°C, aphids produced colonies…thus the absence of aphids on plants used to test the efficiency of Movento at 10°C is linked to unexpected (and unidentified) event.

Why wasn't a full ANOVA factorial analysis done, including main effects and interactions?  The design includes main effects of Host Plant, Treatment, and Temperature, as well as the interactions.  Even if host plants are analyzed separately, a Trt. x Temp interaction is implied by the design for each host plant, but was not reported.  Statistical analysis is not well explained in 4.5.

The experiment has been designed to assess the overall effect of Movento® treatment on infection rates regardless temperature and time after treatment. Thus, statistical analyses carried out and presented  seems consistent with the aim of the experimental design. Also considering the effective size of modalities (10 and 6 plants for treated and untreated modalities, respectively), the introduction of DAT, temperature and their interactions with or without treatment factor is not adapted. Indeed, even if there is an effect, variations induced by the presence of a putative false positive or negative is substantial (10% to 16.7 % for treated and untreated plants). Considering these observations, it is not consistent to include both time and temperature factors in the analysis of the produced data set.

The 4.5 section has been edited.

A key question is whether virus transmission and infection in the host plants was significantly reduced by the Movento treatments.  Although Fig. 3 shows differences between treated vs. untreated plants, even the treated plants had infection rates of ca. 60%.  This would not appear to be a significant advantage in terms of overall virus epidemiology or applied field conditions, but needs to be discussed in terms of expected impacts on yield.  It is hard to imagine that any benefits in yield would be obtained.

In addition to its effect on aphid biology, Movento® reduces by 10% the transmission rate when it is applied preventively. This effect is significant for both plants species when all the data is combined regardless the DAT and the temperature. Statistical parameters have been added in the text and on the figure to clarify our purpose. As in-field insect-transmitted viral disease spread is mainly due to secondary infections, it seems that the increased aphid mortality and the decrease of aphid fecundity following Movento® spray would have the strongest effect on the epidemiology of TuYV and BMYV. Moreover, by reducing (even slightly) the primary infections, less infected plants will be available as viral sources for the vector to acquire and spread the virus in the field. Then, we might suppose that disease spread temporal dynamic will be slow down and could even lead to maintain a higher proportion of healthy plants at the end of the aphid/virus exposure period. Our experiments do not assess these hypotheses. However, we do not have the possibility to quantify how this epidemiological parameter (efficiency of primary infection) will impact virus epidemiology/plant yield. Precisely, we said that “This decrease of the number of infected plants contribute to the improvement of the sanitary quality of the protected areas by reducing the number of virus introductions in the field”. Then, we provide information on the fact that insects-borne diseases mainly rely on secondary infections to accomplish their spread “In studies carried out on other annual plant species (e.g. cereals, potato), it has been shown that the overall incidence of insect-transmitted viruses relies mainly on the efficiency of secondary infections initiated from the primary infected plants [32,33].”. Finally, we suggest that by disturbing aphid biology and transmission process, Movento® may have an effect on diseases spread “Thus, by preventing aphids from settling on treated plants and producing their offspring, and by lowering the success of infection success, Movento® may negatively impacts important steps of viral disease spread at the field level.”.

To moderate our statements, the text of the manuscript has been edited.

Reviewer 2 Report

Efficiency and persistence of Movento® treatment against Myzus persicae and the transmission of aphid-borne viruses

 General comments:

Armand et al., tested the impact of Movento® on Myzus persicae and the aphid-mediated virus transmission under different growing temperatures on rapeseed and sugar beet plants. The results showed that Movento® provided rapeseed and sugar beet from aphid infections and decreased (~10%) the aphid-mediated viral transmission. Other than that, they also tested temperatures on the persistence of aphicide properties. In general, the writing is clear and easy to follow. However, I worry that Movento® is a commercial pesticide, and have been shown in protecting other crops from the green peach aphids. In this sense, no significant novel information is provided. Furthermore, the data presentation need to be significantly improved to increase clarity and reduce confusions. I will leave to the editors for decisions on the suitability for its publication in Plants.

(There are no line numbers or page numbers in the manuscript, thus I will just list some comments in general):

Major comments:

  1. In the introduction, previous studies about Movento® on aphids needs to be explicitly presented. What differences are between this study and the previous studies and what new information the current study present comparing to previous studies?
  2. For all figures, indications on the corresponding statistics will be helpful for readers to easily catch the point of the figures.
  3. Figure 1A, error bars are missing, my understanding is that the experiment was only performed once, and at least the variations across different plants need to be presented.
  4. Figure 2A, each symbol represents data from a single plant? It is not entirely clear what each circle or triangle represents?
  5. Figure 2B, there was no control data presented. Without controls, some of the statement in the manuscript is not accurate. For example, the author stated that “Full aphicid protection last at least 25 days (for the 6 and 10 degree conditions)…”, as the data from figure 1, in most of the cases, no aphids survival even on the non-treated plants at those temperatures. Thus, it is not convincing that the protection is coming from Movento®.
  6. Table 1: the data presented is rather preliminary. At least a conversion of the raw data into percentages to allow direct comparisons between different groups.
  7. Figure 3: The ELISA data needs to be presented. In addition, the statistics for the difference test needs to be presented. To my understanding, the differences in the virus transmission is mainly caused by the decreases in aphid, but not a direct cause from Movento® to virus infections, further discussion may be needed for this possibility.
  8. Figure 4: in the choice assays, it is just confusing by leaving out what does the “both plants” and “any plants” mean. And why can’t replace the “left vs Right plants” with “treated vs non-treated”? To be able to understand why one aphid could be found in Both plants or any plants, which confused me, more details probably need to be provided for how the choice assays were done.
  9. Figure 5: the data presented seems about the percentages of plants were colonized by aphids, however, the description is about aphid survival. These are two totally different things. Either the data needs to be replaced with correct data, or the explanation needs to be corrected to reflect what the data says.
  10. I also caught some typos or incomplete and/or confusing sentences, format inconsistencies, the authors need a read through for those minor things to be corrected.

Author Response

(The authors gave the same response as above.)

Reviewer 3 Report

The study is mostly well designed and obtained results that are important to publish.  However, the authors would do well to clarify a few aspects of the design, analysis, results and conclusions.

  1. If temperatures of 10oC or below are rare, why were levels corresponding to 6o and 10o  included in the design?  Once included in the design, why were they only tested for sugarbeets and not rapeseed?
  2. Figure 1: why didn't aphids establish on sugarbeet at 10o, but did establish at 6o or above 10o?
  3. Why wasn't a full ANOVA factorial analysis done, including main effects and interactions?  The design includes main effects of Host Plant, Treatment, and Temperature, as well as the interactions.  Even if host plants are analyzed separately, a Trt. x Temp interaction is implied by the design for each host plant, but was not reported.  Statistical analysis is not well explained in 4.5.
  4. A key question is whether virus transmission and infection in the host plants was significantly reduced by the Movento treatments.  Although Fig. 3 shows differences between treated vs. untreated plants, even the treated plants had infection rates of ca. 60%.  This would not appear to be a significant advantage in terms of overall virus epidemiology or applied field conditions, but needs to be discussed in terms of expected impacts on yield.  It is hard to imagine that any benefits in yield would be obtained.

Author Response

(The authors gave the same response as above.)

Round 2

Reviewer 2 Report

I don't have any further comments and suggestions for the content.